Review Article

# Metabolic reprogramming in polycystic kidney disease and other renal ciliopathies

Sara Clerici [ID] & Alessandra Boletta [ID] ✉

## Abstract

Primary cilia are solitary organelles formed by a microtubule-based skeleton protruding in a single copy on the surface of most cells. Alterations in their function cause a plethora of human conditions collectively called the ciliopathies. The kidney is frequently and severely affected in the ciliopathies, presenting with a spectrum of phenotypes. Cyst formation is a common trait of all renal ciliopathies. Besides this common manifestation, however, the renal ciliopathies present with profoundly different phenotypes, resulting in either polycystic kidney disease (PKD) or nephronophthisis (NPH) phenotypes. The past decade has seen a surge of studies highlighting metabolic reprogramming as a major feature of PKD, with a distinct involvement of mitochondrial dysfunction. This discovery has brought forward the development of novel therapeutic approaches. More recent evidence suggests that primary cilia modulate the mitochondrial production of energy in response to environmental cues. Here, we summarize the evidence available to date and propose a more general involvement of metabolic and mitochondrial alterations in the renal ciliopathies that might in principle help defining the profoundly different, and potentially opposite, manifestations observed.

**Keywords** Cilia; Kidney Cysts; Ciliopathies; Metabolism; Mitochondria
**Subject Categories** Metabolism; Urogenital System

## Structure of primary cilia

The primary cilium is a microtubules-based organelle protruding as a single copy from most cell types in the human body (Reiter and Leroux, 2017). In polarized cells, such as in epithelia or endothelia, the primary cilium extends from the apical side of the membrane considerably over the microvilli. Despite its apparent simplicity, building, maintaining and functioning of this organelle is extremely complex. For detailed description, we refer to recent reviews (Anvarian et al, 2019; Gopalakrishnan et al, 2023). In brief, ciliogenesis initiates when the mother centriole of a non-dividing cell migrates towards the plasma membrane (on the apical side in polarized cells) to become the basal body of the nascent cilium, and it starts extending a microtubule-based structure, called the axoneme (Reiter and Leroux, 2017; Anvarian et al, 2019). The axoneme is composed of a central microtubule core (typically $9 + 0$) that serves as a rail along which "trains" composed of intraflagellar transport proteins (IFT) move up (towards the tip) or down (towards the base) using kinesin-2 or dynein-2 as motor proteins, respectively, to deliver or retrieve their cargo (Fig. 1). The cilium is a peculiar organelle because its membrane is in continuity with that of the cell and yet the composition of proteins and lipids is highly specialized. This compartmentalization is achieved by a structure located above the basal body, called the transition zone, where Y-shaped structures (the transition fibers) are assembled to create a gating filter for proteins that can enter the axoneme, with a cut-off of ≈70 kDa, and by the assembly of the IFT trains carrying specific cargoes at the base of the cilium (Reiter and Leroux, 2017; Anvarian et al, 2019) (Fig. 1).

## Function of primary cilia

Primary cilia exert different functions in different organs. In general, they act as reservoirs of receptors or channels whose activity is to receive information from the extracellular environment. Activation of ciliary receptors could occur through chemical ligands, such as chemokines or growth factors, or through mechanical signals such as changes in fluid flow causing bending of the cilium, or changes in membrane rigidity (Reiter and Leroux, 2017; Anvarian et al, 2019).

The intracellular signals regulated in response to ciliary receptor activation can be of the most diverse type, aligned with the different functions of tissues. The best-characterized signaling pathway regulated by cilia is that of Sonic hedgehog (Shh) that during development relies entirely on the presence of this organelle for its activation (Bangs and Anderson, 2017). Given the role of cilia as sensors of the extracellular environment it is not surprising to note that this organelle is populated by multiple G-protein coupled receptors (GPCRs) several of which await identification of the ligand, and that it represents a specialized subdomain for GPCR activation (Truong et al, 2021; Hansen et al, 2022). Finally, the best-characterized function of cilia as mechanical sensors of fluid flow is in the node of the developing embryo, where they are involved in establishment of left-right asymmetry during patterning of the mammalian embryo (Gopalakrishnan et al, 2023). A similar function was observed in the kidney epithelium, leading to the proposal of cilia acting as mechanical sensors of flow in the kidney, though the functional outcome of this activity in vivo still awaits to be identified.

Division of Genetics and Cell Biology, IRCCS San Raffaele Scientific Institute, Milan, Italy. ✉E-mail: boletta.alessandra@hsr.it

**Glossary**

**Kidney tubule**
the tubular epithelial structure, regulating fluid composition by secretion and resorption in glomeruli-filtered blood, and producing urines.
**Cyst**
round-shaped fluid-filled structure, formed by epithelial cells visible in pathological conditions in multiple tissues.
**Primary cilium**
single, tiny and immotile organelle made of microtubules protruding as an antenna from multiple cell types.

**Central carbon metabolism (CCM)**
network of catalytic reactions utilizing carbon-based biomolecules to produce energy and/or building blocks in the cells.
**TCA cycle**
a metabolic process occurring in mitochondria, catalyzing carbon-based reactions that release the electrons whose energy is converted into ATP molecules.
**Mitochondria**
multi-functional organelles where the TCA and CCM take place, maintaining cell survival and energy production.

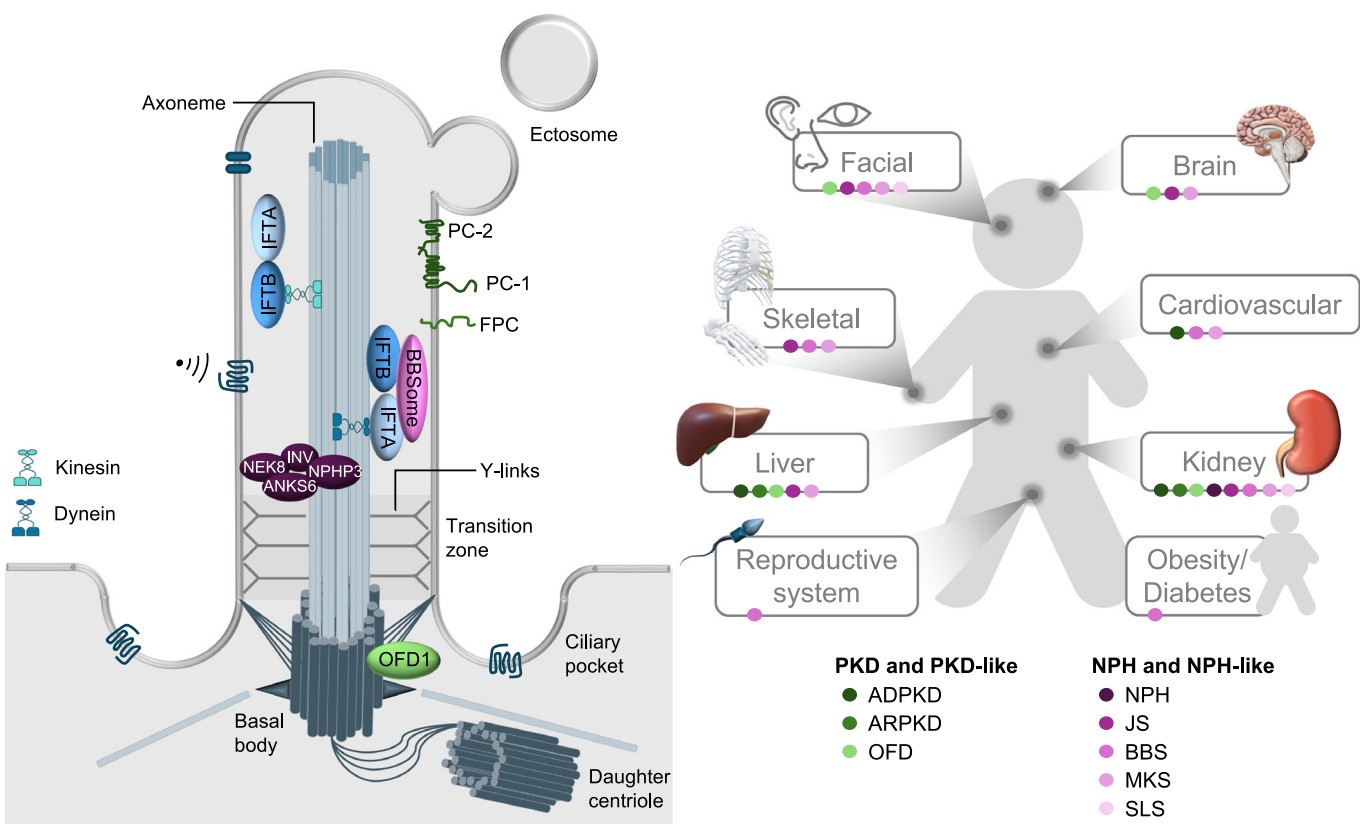

**Figure 1.  Primary cilia and human ciliopathies/renal ciliopathies.**

Left: Schematic overview of ciliary structure. Primary cilia are made of a central microtubule structure that originates from the basal body, initiated by the mother centriole binding to the plasma membrane. Intraflagellar transport molecules and proteins encoded by the genes most frequently mutated in the renal ciliopathies are shown. Right: Schematic overview of the renal ciliopathies and their pleiotropic manifestations. Two major renal phenotypes are observed in the renal ciliopathies, PKD or NPH. Each of the renal ciliopathies presents with alterations in other organs.

# The ciliopathies

Alterations in the function of motile or primary cilia cause numerous different inherited syndromes (>35), each characterized by various phenotypes affecting one or multiple organs, in line with the broad distribution of cilia. The ciliopathies can be classified as motile ciliopathies, when they are caused by disruption of motile cilia and sensory ciliopathies, when they are caused by alterations in the capability of primary, non-motile, cilia to sense external stimuli and to convert them into intracellular biochemical responses (Reiter and Leroux, 2017). The kidney is frequently and severely affected in the sensory ciliopathies, presenting with a spectrum of phenotypes compromising the kidney tubule, which include cyst formation, inflammation, and fibrosis to a variable extent in the different syndromes (see below) (Reiter and Leroux, 2017; Anvarian et al, 2019).

# Renal ciliopathies

Sensory ciliopathies are characterized by manifestations in multiple organs, and renal ciliopathies do not make an exception. Indeed,

each of the different renal ciliopathies reported to date is syndromic, i.e., associated with manifestations in multiple organs (Fig. 1). Here, we will refer to the renal ciliopathies to indicate all the pleiotropic ciliopathies where a kidney manifestation can be observed either as a primary manifestation (i.e., leading to declining kidney function), or as a secondary manifestation (i.e., patients have primarily a dysfunction in an organ different than the kidney, with some involvement of a renal manifestation). In general, cyst formation is a common trait of all these disorders (Fig. 1). Cysts are enclosed epithelial structures that manifest as ballooning or fusiform dilatations of the kidney tubule. In addition to cyst formation, inflammation, and fibrosis are shared by all conditions. Of note, some human conditions prevalently manifest with ballooning of cysts that cause major volume enlargement of the kidney over time. Inflammation and fibrosis are observed in these conditions but appear to be late events (Bergmann et al, 2018). In contrast, some renal ciliopathies manifest with predominant and precocious inflammation and fibrosis, with minimally cystic tissue. These last conditions typically are characterized by smaller kidneys (Yoder, 2007; Arts and Knoers, 2013; Reiter and Leroux, 2017; Devlin and Sayer, 2019). Indeed, when thinking about ciliopathies that are primarily characterized by kidney impairment two major classes can be identified: Polycystic Kidney Disease (ADPKD and ARPKD) (Bergmann et al, 2018) and Nephronophthisis (NPH) (Petzold et al, 2023). Notably, also when kidney involvement is a secondary event in a systemic syndrome, kidney manifestations are either PKD or NPH-like (Arts and Knoers, 2013). This might suggest that different, possibly opposite, ciliary alterations under-line specific pathophysiological mechanisms and robust data from murine models seem to support this notion (see below). An alternative interpretation might be that non-ciliary functions of the proteins encoded by disease genes, or increased severity caused by impaired centrosomal biogenesis might play a role in the kidney manifestation (Cheng et al, 2023).

## Polycystic kidney disease (PKD) and PKD-like

Autosomal Dominant Polycystic Kidney Disease (ADPKD) man-ifests with bilateral kidney cysts, that grow over time eventually causing kidney failure. The second most affected organ is the liver, presenting with massive cyst formation in 80% of cases. Typical ADPKD results from disease-causing variants in either the *PKD1* or *PKD2* genes, encoding for the polycystins (PC1 and PC2, respectively). ADPKD is a two-hit disease, with the normally inherited allele undergoing somatic mutation and leading to cysts formation. Stepwise events in cyst formation include morphological changes of the epithelium, increased proliferation, increased fluid secretion, and matrix remodeling (Yoder, 2007; Ong and Harris, 2015; Bergmann et al, 2018). Only one compound was approved as specific ADPKD therapy to date (Torres et al, 2025).

The precise mechanism of cyst formation as well as the function of the polycystins remain largely elusive to date (Boletta and Caplan, 2025). The polycystins form a membrane-bound receptor/ channel complex whose localization has been observed in multiple subcellular compartments, besides the primary cilium, the cell–cell/ matrix interface and, more recently, the mitochondria–ER contact sites (MERCs, formerly MAMs). Finally, the C-terminal domain

can translocate either into the nucleus or into mitochondria. The complexity behind the biology of the polycystins likely explains the persistent gap in understanding their function (see below) (Padovano et al, 2018; Boletta and Caplan, 2025).

Autosomal Recessive Polycystic Kidney Disease (ARPKD) is a rare disease affecting 1:20,000 individuals, with a most prominent manifestation in infancy. Despite the similarities in the two PKD disorders, the insurgence and progression of ARPKD is profoundly different from that of ADPKD (Bergmann et al, 2018; Burgmaier et al, 2025). Recessive PKD is mostly diagnosed in utero or during perinatal life, and manifests mainly in childhood. Primary clinical manifestation is kidney cyst formation and organomegaly. Furthermore, affected individuals are born with congenital hepatic fibrosis and manifest dilated bile ducts and portal hypertension (Burgmaier et al, 2025).

ARPKD is caused by homozygous mutation in the Polycystic Kidney and Hepatic Disease 1 (*PKHD1*) gene. *PKHD1* encodes for a large transmembrane protein called Fibrocystin/Polyductin (FPC), of unclear function and localized in primary cilia. In addition to its localization into the primary cilium, cleavage products of FPC were found in the cytosol, in the nucleus and into mitochondria (see below) (Walker et al, 2023).

Multiple genetic conditions have recently been described as "atypical" PKD referring to the fact that subsets of patients manifest with a similar phenotype as ADPKD, but much milder and clinically distinguishable. Among the several genes found mutated in these atypical cases are *IFT140*, *NEK8*, *GANAB*, *ALG9*, *ALG5*, *DNAJB11* (Torres et al, 2025).

Finally, among the other renal ciliopathies that manifest with a PKD-like phenotype is Oral-facial-digital syndrome type 1 (OFD1), a rare X-linked dominant ciliopathy associated with congenital malformations, including cleft palate, cognitive impairment, and digital anomalies. Kidney manifestations are observed in 50% of the patients, typically manifesting with multicystic kidney disease, resulting in increased kidney volume. Cysts can also be observed in liver and pancreas, similarly to typical ADPKD (Arts and Knoers, 2013; Devlin and Sayer, 2019).

## Nephronophthisis (NPH) and NPH-like

NPH represents the most common genetic disorder causing kidney failure in infants and adolescents. Clinically, it is characterized by polyuria and polydipsia due to reducing urinary concentrating ability, followed by tubulointerstitial fibrosis and formation of cysts at the cortico-medullary junction. However, great variability can be observed in this disorder. It is typically classified as infantile, or juvenile/adult based on the age of onset. Typically, early onset cases manifest in utero and infants reach kidney failure by the age of three. This is the rarest form of NPH presenting with moderately enlarged kidneys and cortical cysts (Arts and Knoers, 2013; Devlin and Sayer, 2019). On the contrary, the most common form of NPH presents as juvenile or adult-onset. In this case disease manifests at a young age or in adults and presents with a slightly different phenotype manifesting typically with polydipsia and polyuria, with small to normal-sized kidneys, reduced cortico-medullary differentiation and the presence of cortico-medullary cysts. Thus, while cysts can be observed in both infantile and juvenile NPH, the size of the kidney points to a slightly larger organ in infants and a reduced-size kidney in juvenile and adult patients, with the notable exception of cases with grossly enlarged

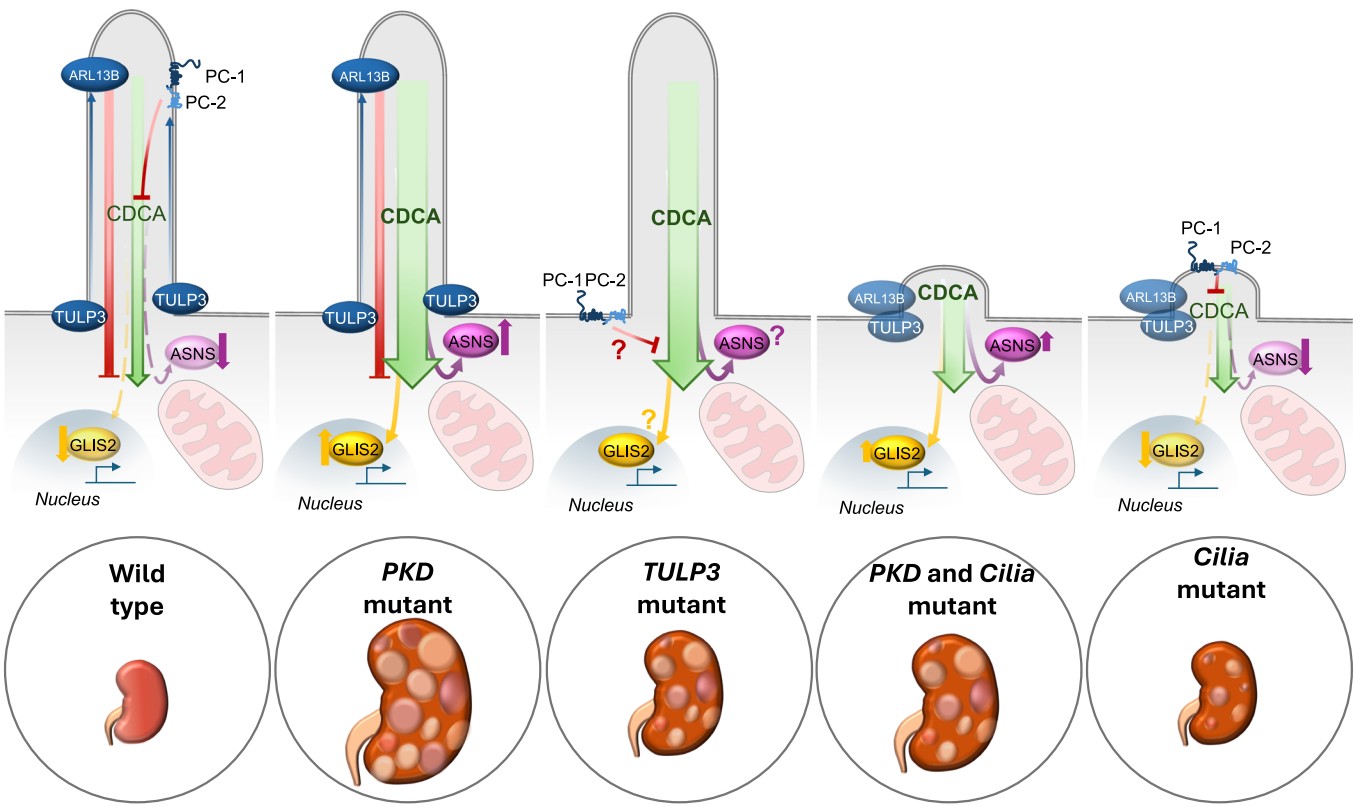

**Figure 2. Counteregulatory pathways driving cystogenesis from cilia.**

Schematic representation of the renal manifestations observed in mutants known to influence a PKD phenotype. Green arrow represents the Ciliary Dependent Cyst Activation (CDCA) mechanism, regulated by the polycystins. Red arrow represents the inhibition of a cilia-derived minor cystogenic signal by ARL13B. TULP3 acts as a gatekeeper for ciliary localization of the polycystins and ARL13B. ASNS and GLIS2 are potential downstream players of CDCA signal.

kidneys resembling ARPKD (Arts and Knoers, 2013; Devlin and Sayer, 2019; Halawi et al, 2023).

In sum, NPH has morphological presentation characterized by hypoplastic kidneys, increased inflammation and fibrosis and cortico-medullary cysts, with the exception of the infantile NPH which might correlate with slightly enlarged kidneys.

Among the disorders that manifest with NPH-like disease, the most frequent are the syndromic ciliopathies Joubert Syndrome (JS) and Bardet–Biedl Syndrome (BBS), Jeune syndrome, Meckel-Gruber Syndrome, and Senior-Loken syndrome. Not only can these disorders manifest with kidney phenotypes observed in NPH, but they also can share disease-causing variants in the same genes, though resulting in manifestations that differ from that of isolated NPH, for reasons not yet fully understood (Mykytyn et al, 2001; Arts and Knoers, 2013; Devlin and Sayer, 2019).

## Primary cilia in the kidney: the Yin and Yang of cell growth

Work from animal models has unveiled a quite complicated relationship between cilia function and kidney cystogenesis. Inactivation of any ciliary gene, such as the intraflagellar transport 88 protein (*Ift88*) or the ciliary kinesin *Kif3a*, in kidney epithelia results in cystogenesis. However, the phenotype of *Ift88* or *Kif3a*

mutants is much milder and delayed as compared to cyst formation observed upon inactivation of the *Pkd1* or *Pkd2* genes (Ma et al, 2013). Furthermore, when very aggressive cystogenesis is induced by inactivation of the *Pkd1* and *Pkd2* genes, then concomitant removal of cilia (by inactivation of the same key ciliary genes *Ift88* or *Kif3a*) has a strikingly protective effect, with kidney cystogenesis being drastically delayed (Ma et al, 2013). The proposed mechanism that explains these results is that the receptor-channel complex of the polycystins at ciliary membrane inhibits an important pro-cystogenic pathway that requires the presence of the ciliary microdomain to be efficiently activated (Fig. 2). This signal has been named CDCA (Cilia-Dependent Cyst Activating signal) (Ma et al, 2013; Ko and Park, 2013; Shao et al, 2020; Ma, 2021). The precise molecular details of this pathway remain to be defined, with a few components being identified. The ciliary gatekeeper TULP3 regulates trafficking of the polycystins to cilia impacting on the CDCA or a parallel pathway (Fig. 2) (Legué and Liem, 2019; Ma, 2021). The transcription factor GLIS2 was also shown to be a CDCA component, since its activity is elevated in *Pkd1* and *2* mutants, removal of cilia reduces its activity, and its inhibition ameliorates disease progression. Of interest, the gene *GLIS2*, when mutated in homozygosity in humans, gives rise to nephronophthisis, in line with the idea that some ciliary alterations lead to increased cell growth, while others lead to the opposite manifestation, i.e., reduced growth (Zhang et al, 2024). In line with

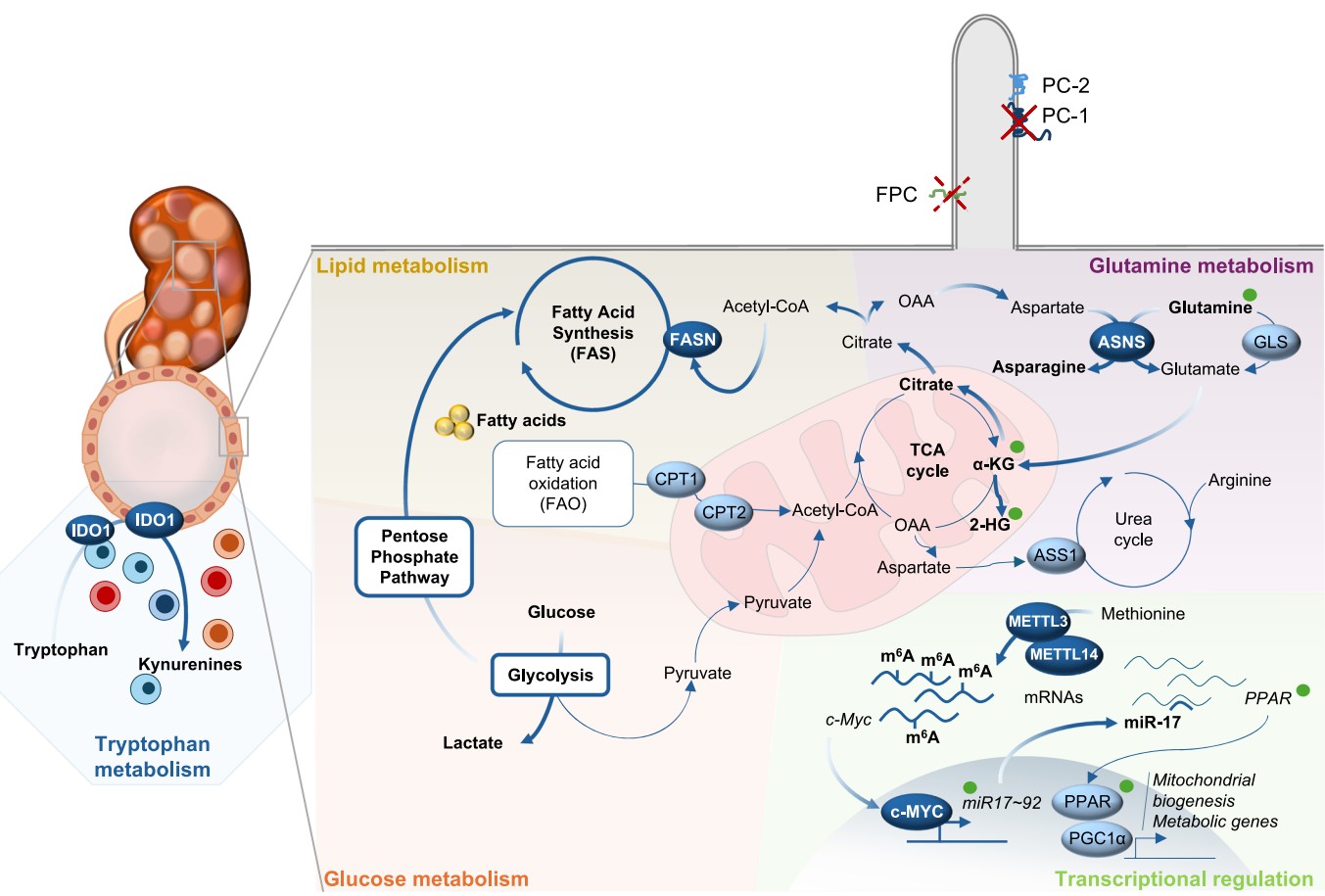

**Figure 3. Metabolic reprogramming in PKD.**

Cartoon summarizing the principal metabolic pathways deregulated in Autosomal Dominant Polycystic Kidney Disease, including glycolysis, glutamine anaplerosis, TCA cycle, OXPHOS and other central carbon metabolism alterations. The few alterations observed in recessive PKD (ARPKD) are indicated by green dots. Additional metabolic alterations influence the cystic microenvironment and the immune response.

the above, *Glis2* mutant mice develop NPH (Lu et al, 2016). Similarly, the metabolic enzyme ASNS was shown to be elevated in *Pkd1* mutant kidneys, is reduced in cells lacking cilia, and its inhibition retards disease progression in *Pkd1* mutant mice (Podrini et al, 2018; Steidl et al, 2023; Clerici et al, 2024) (Fig. 2). Additional components of the pathway remain to be defined, but what we gather from these two components of the CDCA pathway is that they seem to be potent drivers of proliferation/growth. This is interesting because, even if multiple studies demonstrated that increased proliferation per se is not sufficient to drive cystogenesis (Sharma et al, 2013; Pema et al, 2016), and the rate of proliferation appears to be slow in PKD human and murine samples, the relentless increase of proliferation and growth is certainly driving volume increase in PKD (Boletta and Caplan, 2025).

## Metabolic reprogramming in polycystic kidney disease

Indeed, multiple pro-proliferative signaling pathways were found altered in PKD and have been targeted for therapy (Boletta and Caplan, 2025). In the past decade metabolic reprogramming also

emerged as a major driver in ADPKD. The first observation came from evidence that glucose utilization represents the major source of energy in ADPKD cells and tissues (Rowe et al, 2013). Indeed, PKD cells derive their energy from the so-called "aerobic glycolysis" also known as the "Warburg effect" oxidizing pyruvate to lactate (Rowe et al, 2013; Podrini et al, 2018) (Fig. 3).

Further analysis of animal models and human tissues of ADPKD confirmed the existence of a strong glycolytic signature (Lian et al, 2017; Podrini et al, 2018; Baliga et al, 2021; Clerici et al, 2024). Definitive validation came by in vitro and in vivo tracing studies with $^{13}C_6$-glucose in PKD models (Rowe et al, 2013; Chiaravalli et al, 2016; Podrini et al, 2018). The sensitivity of this technology allowed to resolve the previously debated evidence of an enhanced glycolytic flux in PKD (Warner et al, 2016; Menezes et al, 2016). Notably, glucose was also found to boost cyst growth in vitro (Kraus et al, 2016) and in human iPS-derived organoids (Li et al, 2022). Glucose deprivation strongly impairs cell growth in PKD models (Rowe et al, 2013; Podrini et al, 2018; Li et al, 2022). In line with this, treatment with the glucose analog 2-deoxy-D-glucose (2DG) improves disease progression in preclinical models of PKD (Rowe et al, 2013; Chiaravalli et al, 2016; Riwanto et al, 2016; Lian et al, 2019; Atwood et al, 2024). Hyperglycemia and disease

worsening correlated in a small cohort of PKD patients and ketosis retarded disease progression in animal models (Torres et al, 2019).

Further studies showed that while glucose minimally fuels the TCA cycle resulting in reduced OXPHOS, PKD cells compensate with glutamine utilization (Rowe et al, 2013; Podrini et al, 2018) (Fig. 3). Tracing studies using $^{13}C_5$-glutamine revealed that the majority of glutamine is diverted towards the production of m + 5 citrate through reductive carboxylation to fuel fatty acids biosynthesis (Podrini et al, 2018) (Fig. 3). Several studies demonstrated the centrality of glutamine in ADPKD progression (Flowers et al, 2018; Podrini et al, 2018; Soomro et al, 2018). Some studies proposed that the enzyme glutaminase (GLS-1) drives glutaminolysis in PKD, similar to cancer (Flowers et al, 2018; Soomro et al, 2018). An alternative mechanism proposed to drive glutaminolysis in PKD was via the enzyme asparagine synthetase (ASNS). Indeed, ASNS is upregulated in cystic kidneys of orthologous PKD animal models and in different datasets of ADPKD human samples (Podrini et al, 2018; Clerici et al, 2024), and Asns silencing in vitro (Podrini et al, 2018), and in vivo (ASO) resulted in a strong amelioration of the PKD phenotype (Clerici et al, 2024). Coherently, children and young adults diagnosed with ADPKD feature an increase in circulating asparagine which correlates with disease severity (Baliga et al, 2021).

Other amino acids were also shown to be important for PKD progression. Cystic tissues, undergoing loss of argininosuccinate synthase-1 (ASS1) during the progression of the phenotype, develop exogenous arginine dependence, and when arginine is not available, glutamine provides a compensatory mechanism (Trott et al, 2018). Methionine metabolism, important for $N^6$-methyladenosine RNA modification and targeting of METTL3 or methionine dietary restriction were shown to ameliorate PKD (Ramalingam et al, 2021). Increased tryptophan metabolism was associated with an immunomodulatory effect in ADPKD (Baliga et al, 2021; Nguyen et al, 2022).

Finally, lipid metabolism is profoundly affected in PKD (Menezes et al, 2012, 2016), with impaired FAO and lipid utilization to fuel OXPHOS in Pkd1 mutant cells and tissues (Fig. 3) (Song et al, 2009; Menezes et al, 2016; Hajarnis et al, 2017; Lakhia et al, 2018; Podrini et al, 2018; Lin et al, 2018; Daneshgar et al, 2021).

Among the mechanisms proposed to deregulate metabolism, miR-17 upregulation seems to play an important role (Patel et al, 2013; Hajarnis et al, 2017; Yheskel et al, 2019), and likewise the PPAR family of transcription factors (Hajarnis et al, 2017; Lakhia et al, 2018) (Fig. 3).

Multiple lines of evidence also reported metabolic reprogramming in ARPKD, showing alterations aligned with those of ADPKD. Indeed, ARPKD mouse models and patient-derived tissues show a dependence on glutamine (Hwang et al, 2015). This corroborates the crucial role of glutamine in cystic disorders, sustaining proliferation in both dominant and recessive forms of PKD (Fig. 3). Similarly, alterations in key regulators of lipid metabolism and upregulation of miR-17 are observed also in ARPKD (Patel et al, 2013; Hajarnis et al, 2017; Yheskel et al, 2019), and treatment with pioglitazone and rosiglitazone, activating PPARγ improved kidney and liver phenotypes of orthologous rat models of ARPKD (Blazer-Yost et al, 2010; Yoshihara et al, 2011). Finally, the recent discovery that OXPHOS is reduced in the absence of the Pkhd1 gene due to a direct impact of the fibrocystin/

polyducting protein on mitochondria (Walker et al, 2023) represents a major breakthrough and calls for further investigations on possible metabolic alterations in this disease as therapeutic targets.

# Central role of mitochondria in PKD

Given the radical changes in metabolic regulation and bioenergetic pathways, the possible role of mitochondria in PKD and/or other renal ciliopathies were undertaken. Indeed, defective OXPHOS in ADPKD (Song et al, 2009; Menezes et al, 2012), validated in multiple murine and human tissues settings (Muto et al, 2022) lead to the consistent observation that Pkd1-deficient cells show decreased oxygen consumption rate (OCR) (Padovano et al, 2017; Ishimoto et al, 2017; Podrini et al, 2018). A single study has shown by in situ measurement of cytochrome c oxidase (COX) and succinate dehydrogenase (SDH) that mitochondrial respiration is impaired in the cystic epithelia of PKD mouse kidneys (Cassina et al, 2020). Functional impairment was paralleled by alterations in mitochondrial morphology, with fragmented and swollen mitochondria showing defective cristae being reported in Pkd1 mutants (Ishimoto et al, 2017; Lin et al, 2018; Cassina et al, 2020), correlating with the accumulation of reactive oxygen species (ROS) and oxidative stress (Ishimoto et al, 2017; Padovano et al, 2017; Kahveci et al, 2020).

In line with these lines of evidence, rescuing mitochondrial biogenesis (Hajarnis et al, 2017; Yheskel et al, 2019; Lee et al, 2019), restoring the unbalanced mitochondrial dynamics (Cassina et al, 2020), or overexpression of mitochondria-targeted catalase (mCAT) all resulted in amelioration of the PKD phenotype (Lu et al, 2020; Daneshgar et al, 2021). Worth mentioning here, Fedeles et al proposed the opposite therapeutic approach based on exacerbating oxidative stress as a way to unbalance and kill cells mutant for the Pkd1 gene by using 11beta-dichloro treatment (Fedeles et al, 2024).

One important question that emerged from the studies above is how central and important is mitochondrial dysfunction in the initiation of the disease? Of note, multiple studies point toward a direct involvement of the ADPKD mutated proteins, the polycystins, or the ARPKD mutated protein, FPC, in regulation of mitochondria physiology. First, PC1 and PC2 were found to localize at mitochondria-associated ER membranes (MERCS, formerly MAMs), impacting on $Ca^{2+}$ influx and regulation of OXPHOS (Padovano et al, 2017).

A second mechanism proposed on how the Polycystins might regulate mitochondrial activity, came from evidence that a hidden Mitochondrial Targeting Sequence (MTS) becomes recognizable upon cleavage of the C-terminal fragment (CTT) leading to its import into mitochondria (Lin et al, 2018) (Fig. 4). In line with this, overexpression of PC1-CTT ameliorates the cystic phenotype and kidney function in Pkd1 mutant mice, by interacting with and regulating the activity of the mitochondrial enzyme Nicotinamide Nucleotide Transhydroxygenase (NNT) (Onuchic et al, 2023).

As indicated above, recent data demonstrated that Fibrocystin, the PKHD1 gene product, also releases its C-terminal tail, unveiling a Mitochondria Targeting Sequence, which drives import into mitochondria. FPC loss in cells impairs mitochondrial morphology

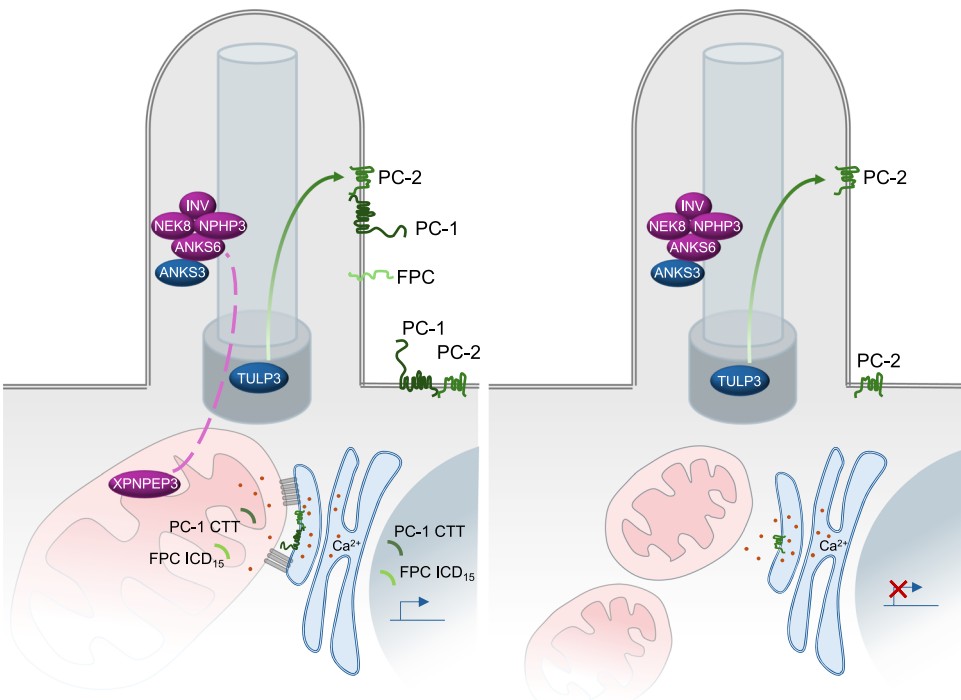

**Figure 4. Mitochondrial translocation of PKD & NPH proteins.**

Left: The Polycystins and Fibrocystin were linked to mitochondria regulation, by direct physical translocation of small fragments into the mitochondria and/or regulation of Mitochondria–ER membrane structures (MERCs). Limited information links NPH proteins to the regulation of mitochondria, with one reported gene mutated being exclusively in mitochondria, and with a nephronophthisis complex comprising and interacting with mitochondrial proteins (Hoff et al, 2013). Right: Mitochondria display multiple alterations in PKD, including mitochondria fragmentation and reduced calcium uptake.

and respiration (Chumley et al, 2019), and its C-terminal fragment restores it (Walker et al, 2023) (Fig. 4).

## Metabolic reprogramming and mitochondria involvement in nephronophthisis

Only few studies have been published on possible metabolic alterations in cells invalidated for specific NPH genes. Inactivation of the *Anks6* gene, known as *NPHP16*, in mIMCD3 cells resulted in reduced amino acids consumption and reduced proliferation (Schlimpert et al, 2019). Similarly, inactivation of *Anks3*, a putative NPH gene and an interactor of the *Anks6*, also resulted in changes in metabolic profiling and reduced proliferation rate (Schlimpert et al, 2018). While the generation of animal models of NPH more faithful to the human condition are still underway, several murine models of the disease have been studied. Among these, analysis of the kidneys from the *Jck* mouse, which carries a mutation in the *Nek8* gene, revealed an overall change in purines in the urine of the mice at early stages (Taylor et al, 2010). These data are interesting considering a study that has characterized potential interactors of the *Anks6* gene, identifying the ANKS6-ANKS3-NEK8-INVS-NPHP3 module, and further showing an unexpected interaction with mitochondrial proteins (Hoff et al, 2013) (Fig. 4). Notably, while most of the NPH genes described to date encode for ciliary proteins, there is one notable exception of the gene *XPNPEP3*, mutated in a family affected by an NPHP-like nephropathy and which is localized exclusively in mitochondria (Tong

et al, 2023). These few pieces of evidence suggest that a metabolic alteration might be present in NPH. Besides the reduced proliferation observed in the studies above that might partially explain the smaller kidney phenotype observed in NPH, increased apoptosis was also reported in NPH (Wodarczyk et al, 2010), possibly playing a role. Finally, it should be noted that NPH kidneys display mostly enhanced fibrosis. Since there is extensive literature linking metabolic derangement with kidney fibrosis (Chung et al, 2019; Pu et al, 2024; Miguel et al, 2025), it is possible that mitochondrial and metabolic reprogramming might impact this aspect of the renal ciliopathies phenotype.

## Primary cilia and metabolic regulation

As highlighted before, the human ciliopathies result in a wide spectrum of manifestations, that are at least in part explained by the broad distribution of cilia in different tissues. Since the first discovery of cilia involvement in human pathology, one fundamental discovery is that cilia regulate systemic metabolism via regulation of hormonal responses in at least two conditions: obesity and type 2 diabetes, both frequently manifesting in some ciliopathies, the most prominent Bardet–Biedl Syndrome (BBS) (Fig. 5). BBS is caused by disease-causing variants in several genes multiple of which encode for proteins that work in a complex called the BBSome, whose function in neuronal cilia in the hypothalamus has been extensively studied (Mykytyn et al, 2001; Oh et al, 2015;

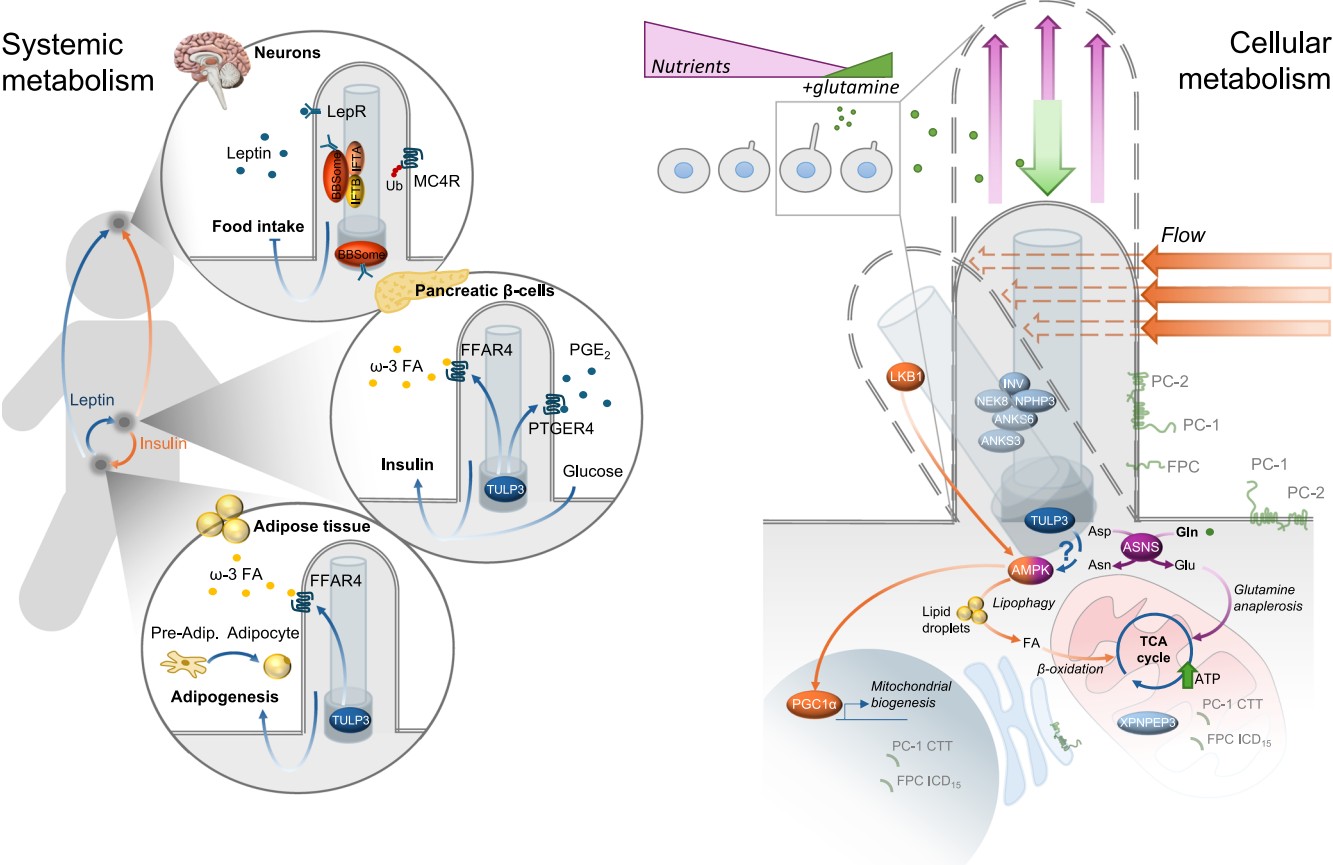

**Figure 5. Primary cilia and regulation of metabolism.**

Left: Primary cilia play a central role in systemic metabolism by responding to various of hormones, including regulation of satiety and insulin release, in line with the high frequency of obesity and type 2 diabetes in the ciliopathies. Right: Recent evidence showed that individual cellular metabolism is regulated by primary cilia. Mechanical bending, or nutrients (glutamine) availability sensed by cilia regulate cellular metabolism. Under nutrient deprivation, when glutamine becomes the obligated source of carbon, the enzyme asparagine synthesase (ASNS) in cilia senses glutamine levels favoring its utilization in mitochondria.

Guo et al, 2016). Patients affected by BBS manifest with obesity, due to improper leptin receptor signaling and hence sensitivity to leptin, caused by a malfunctional BBSome (Fig. 5). Most recent work demonstrated that cilia in pre-adipocytes regulate differentiation to adipocytes in response to omega-3 fatty acids, expanding the mechanistic explanation for obesity in the ciliopathies (Hilgendorf et al, 2019) (Fig. 5). Additionally, several studies demonstrated that primary cilia in pancreatic β cells regulate insulin secretion, allowing control of glycemia (Volta et al, 2019; Hughes et al, 2020) (Fig. 5). Thus, a prominent role for cilia in systemic metabolism might explain the high incidence of obesity and diabetes among patients affected by the ciliopathies.

Further work also demonstrated that primary cilia regulate authophagy in a cell-autonomous manner (Pampliega et al, 2013). Furthermore, mechanical bending of cilia was found to regulate the energy sensor protein AMPK and the activation of the mTORC1 cascade (Boehlke et al, 2010; Miceli et al, 2020) acting on a specific type of autophagy, called lipophagy, and contributing to regulate β-oxidation (Miceli et al, 2020) (Fig. 5). Autophagy and the mitochondrial redox state have also been linked to proper ciliary function in dopamine neurons, leading to hypothesize a role for

primary cilia in the onset of Parkinson's disease (Bae et al, 2019). In a recent study, we uncovered that primary cilia play a fundamental role in the cell-autonomous regulation of bioenergetic pathways, by sensing the availability of nutrients (glutamine in particular) and by responding to nutrient availability by fine-tuning mitochondrial activity and OXPHOS in vitro and in vivo (Steidl et al, 2023). We have shown that this activity is very important during metabolic stress conditions when reduced mitochondrial activity causes an AMPK-dependent elongation of cilia. In these conditions, glutamine, which is a non essential amino acid, becomes conditionally essential and the only carbon source that could be used by mitochondria. The addition of glutamine reverses this effect in a manner mediated by the metabolic enzyme asparagine synthetase (ASNS) (Fig. 5). Of interest, removing cilia from cells reduces their capability to express this enzyme and to utilize glutamine to fuel the TCA cycle and OXPHOS under nutrient stress conditions (Steidl et al, 2023). This is interesting at the light of the dual role of cilia in kidney cystogenesis uncovered by studies on the CDCA pathway and at the light of the role of metabolic reprogramming in supporting cell proliferation in PKD. Of note, while the state of metabolic regulation was never assessed in mice manifesting PKD

due to ablation of TULP3 in the kidney, a recent paper has shown that fasting in the mouse causes the release in circulation of a bile acid called lithocolic acid (LCA) which was shown to directly and physically bind TULP3, which in turn mediates sirtuin-dependent de-acetylation of AMPK leading to its activation (Qu et al, 2024). As mentioned above TULP3 is a central component in building a ciliary structure and in trafficking of multiple receptors at cilia. Given the elongation of cilia observed in the kidney upon fasting in the mouse, further research should be undertaken to understand this possible connection. While these studies are still at the beginning, it would be surprising not to see this new line of investigation flourish in the coming years.

## Concluding remarks

In conclusion, solid multiple evidence support the notion that metabolic alterations are major components of the stepwise events leading to kidney dysfunction in PKD and PKD-like renal ciliopathies. The fundamental role that primary cilia play in the kidney and in the initiation of all renal ciliopathies, along with the recent discovery that cilia can communicate the extracellular availability of nutrients and the environmental energy state to fine-tune the activity of mitochondria and their energy production, clearly calls for extensive studies aimed at understanding the possible involvement of metabolic de-regulation as a central feature of all ciliopathies.

### Pending issue

- Future studies should concentrate on deciphering the molecular mechanisms underlying the ciliary regulation of metabolism in various cell types, including the kidney.
- Studies are required to define the metabolic needs and reprogramming in the various renal ciliopathies caused by mutations in different genes.
- Definition of the role of metabolism in the cilia-dependent cyst activating pathway might lead to innovative understanding of the renal ciliopathies and help define new therapies.

## Peer review information

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

## Acknowledgements

The authors are grateful to other members of the Boletta lab for discussions. Work is supported by the European Research Council (ERCAdG Project 101142691—QtCilia), European Union Horizon Europe (TheRaCiL—Project 101080717), the Italian Association for Research on Cancer (AIRC, IG2024-30769).

## Author contributions

**Sara Clerici**: Visualization; Writing—original draft; Writing—review and editing. **Alessandra Boletta**: Writing—original draft; Writing—review and editing.

## Disclosure and competing interests statement

AB is an inventor of patents related to metabolic interventions in PKD.

