## [Peer Review File · EMBO Molecular Medicine]

Metabolic reprogramming in polycystic kidney disease and other renal ciliopathies

Alessandra Boletta and Sara Clerici

Corresponding author: Alessandra Boletta (boletta.alessandra@hsr.it)

Review Timeline:

Submission Date:	12th Feb 25
Editorial Decision:	3rd Mar 25
Revision Received:	1st Apr 25
Accepted:	4th Apr 25

Editor: Zeljko Durdevic

Transaction Report:

3rd Mar 2025

Dear Dr. Boletta,

Thank you for the submission of your manuscript to EMBO Molecular Medicine. We have now received feedback from the two reviewers who agreed to evaluate your manuscript. As you will see from the reports below, the referees are positive about its interest and timeliness, however, they also raise important criticisms that should be addressed in a revised manuscript.

I would also like to ask you to add the following items to your revised article:

- 1) Please upload figures as high-resolution files in TIFF, EPS or PDF format.
- 2) If BioRender was used to create the figures, please add following sentence to the figure legends: "Graphics were created with BioRender.com."
- 3) Add up to 5 keywords.
- 4) Author contributions: Please remove it from the manuscript and specify author contributions in our submission system. CRediT has replaced the traditional author contributions section because it offers a systematic machine-readable author contributions format that allows for more effective research assessment. You are encouraged to use the free text boxes beneath each contributing author's name to add specific details on the author's contribution. More information is available in our guide to authors:
<https://www.embopress.org/page/journal/17574684/authorguide#authorshipguidelines>
- 5) Please add "Glossary". The glossary is meant to explain some of the terms used for laymen. Could you please identify terms that may need an "explanation"?
- 6) Please add "Pending issue". At the end of each article is a box highlighting issues that still need further studies and where research efforts should converge. Could you identify some pending issues?
- 7) As part of the EMBO Publications transparent editorial process initiative EMBO Molecular Medicine will publish online a Review Process File (RPF) to accompany accepted manuscripts. This file will be published in conjunction with your paper and will include the anonymous referee reports, your point-by-point response and all pertinent correspondence relating to the manuscript. Let us know whether you agree with the publication of the RPF.

I hope that the referees' comments do not prove too problematic to address and I look forward to reading your next version.

Yours sincerely,

Zeljko Durdevic

*** IMPORTANT INFORMATION ***

- 1) a .doc formatted version of the manuscript text (including Figure legends and tables)
- 2) Separate figure files
- 3) a letter INCLUDING the reviewer's reports and your detailed responses to their comments.

Also, and to save some time should your paper be accepted, please read below for additional information regarding some features of our research articles:

- 1) Glossary: EMBO Molecular Medicine articles will be accompanied by a glossary explaining some of the terms used for

laymen. I identified the following:

_____, _____, _____

Could you please help us in identifying terms that may need an "explanation" other terms that we can add to the glossary.

2) For more information: This is a short list of related web links for further consultation by the readers. Could you identify some relevant ones? Examples are patient associations, OMIM related links, databases, authors websites, etc.

3) Pending issues: At the end of each article we will have a box highlighting issues that still need further studies and where research efforts should converge (we call this the Pending issues box). From my reading I would say:

but I can see there may be many more. Could you work on this as well?

4) Disclosure and competing interest statement: Please include a statement declaring any competing commercial interests in relation to your submitted work.

5) Please note that we now mandate that all corresponding authors list an ORCID digital identifier. This takes <90 seconds to complete. We encourage all authors to supply an ORCID identifier, which will be linked to their name for unambiguous name identification.

Currently, our records indicate that the ORCID for your account is 0000-0002-4704-4006.

Link Not Available

-

Thank you,

Zeljko Durdevic

***** Reviewer's comments *****

Referee #1 (Remarks for Author):

This is an interesting and timely review paper connecting cilia structure and function with metabolic reprogramming in various forms of PKD. While the work on metabolic reprogramming in ADPKD is presented in a detailed, accurate, and balanced fashion, the manuscript will benefit from a more in-depth discussion of potential roles of metabolic regulation in nephronophthisis (NPH). As the authors acknowledge, NPH differs from ADPKD, regarding the severity and extent of cystogenesis, fibrosis, and tubular degeneration/death. The authors discuss the connection of several NPH genes (ie Ank6, ANks3, Nek8, and Xpnpep3) in mitochondrial function and cell proliferation, which is appropriate. However, increased proliferation does not always equate to cystic expansion, let alone initiation of cystogenesis. Therefore, caution should be exercised in making this connection. In addition, it would be informative if they extend the discussion to the role of metabolic reprogramming/metabolic regulation in fibrosis and tubular degeneration/death, in the context of NPH. If metabolic reprogramming has a causal role in NPH, it should affect (partially or entirely) fibrosis and/or cell death, as these are the major drivers of kidney function in NPH. In this regard, and as also mentioned above, the discussion of effects of altered mitochondrial metabolism on proliferation in NPH is somewhat superficial, as the reduced kidney size may be due to increased cell death rather than reduced proliferation. The discussion on primary cilia and metabolic regulation is adequate and informative. Overall, this is a well-done manuscript, which I recommend for publication, after expanding a bit the discussion on fibrosis and tubular atrophy/death.

Minor comments:

- 1) P.4 line 2, second paragraph: "receptor activation"
- 2) P.6, line 7, second paragraph: "a two-hit disease"

- 3) P.7, line 5, fourth paragraph: "... genes cause misfolding..."
- 4) P.9, line 12, second paragraph: "...axonemal membrane of cilia..." instead of "plasma membrane".
- 5) P10, lines 17-20, first paragraph: Please add references.
- 6) P.11, line 1, first paragraph. "...plethora of studies have contributed to the delineation of the mechanisms..."
- 7) P.20, line 1, first paragraph, "amino acid..."
- 8) P.20, line 9, first paragraph: "suggest that opposite..."
- 9) P. 18, line 7, first paragraph and P.20, line 1, second paragraph: "...lines of evidence..." instead of "evidences".

Referee #2 (Remarks for Author):

The manuscript by Clerici and Boletta describes recent progress in metabolic programming in polycystic kidney disease and other renal ciliopathies. This is a very well-written manuscript that gives novel insights into a rapidly evolving research field and therefore is of interest for a broader community. It comes timely as recent publications have revealed novel and important aspects. There are a number of aspects remaining that should be addressed prior to consideration for publication.

Major:

- The authors classify ADPKD as a ciliopathy. There is an ongoing debate about this classification. Please consider to rephrase or discuss.
- On page 6 the authors speculate about potential opposite ciliary alterations underlying PKD vs. NPH. Please specify why this could not be alterations in other subcellular organelles.
- The authors state that only one compound was approved for therapy of ADPKD to date. This is not entirely correct. Tolvaptan is the only targeted treatment based on molecular insights but therapy of ADPKD goes far beyond this incl. therapy of arterial hypertension etc. Please rephrase.
- The authors mention that Glis2 responds to all three major characteristics expected from a CDCA component. They mention that removal of cilia reduces Glis2 activity. Yet, loss of Glis2 results in a NPH-like phenotype and loss of Glis2 in Kif3a-mutant mice results in reduction of the Kif3a-phenotype (Lu et al, Kid Int 2016) - does this fit to the concept of the authors? Can they comment?
- On page 17 the authors link variants in Anks6 to cellular metabolism and link this to proliferation. The link remains weak and I understand that causality was not shown. I would support the authors to extend their already cautious wording even further. Can the authors also be more specific here? In the same section it may be important to point out that Glis2 mice show an NPH phenotype.
- The authors speculate that reduced proliferation may be resulting in smaller kidneys. How would the link to fibrosis be in NPH and PKD? Would an additional hit be required?
- Page 18: „BBS is caused by mutations (...) in a complex called the BBSome...". To my knowledge the BBSome includes only some of the proteins that can be affected in BBS. Please double check and comment.
- Please use updated nomenclature e.g. „kidney" instead of „renal", „kidney failure" instead of „ESKD" etc. (<https://pubmed.ncbi.nlm.nih.gov/32409237/> or <https://pubmed.ncbi.nlm.nih.gov/32409780/>), „disease-causing variants" instead of „mutations" etc.

Minor:

- I would suggest to move the description of the different types of cilia (motile vs non-motile, 9+0 vs 9+2) to an earlier part of the manuscript. On page 4 the authors describe sensing at the node of the developing embryo but have not previously properly introduced motile cilia which could be helpful.
- On page 7 the authors state that ARPKD „is diagnosed in utero". While this is true in many cases, there are also more and more emerging descriptions of mild courses of ARPKD. Please rephrase.
- On page 7 the authors cite „Group et al., 2025" - I understand they refer to the KDIGO ADPKD Work group? Please check
- Kidneys in infantile NPH can be very big during early disease and are a differential diagnosis of ARPKD - please consider your statement on page 8.
- It remained unclear to me why Tulp was in the figure 2 until reading further down. Please consider to rearrange the text.

Please find below the comments of the reviewers in black text and our responses in blue text.

***** Reviewer's comments *****

Referee #1 (Remarks for Author):

This is an interesting and timely review paper connecting cilia structure and function with metabolic reprogramming in various forms of PKD. While the work on metabolic reprogramming in ADPKD is presented in a detailed, accurate, and balanced fashion, the manuscript will benefit from a more in-depth discussion of potential roles of metabolic regulation in nephronophthisis (NPH). As the authors acknowledge, NPH differs from ADPKD, regarding the severity and extent of cystogenesis, fibrosis, and tubular degeneration/death. The authors discuss the connection of several NPH genes (ie Ank6, ANks3, Nek8, and Xpnpep3) in mitochondrial function and cell proliferation, which is appropriate. Thanks for the appreciation.

However, increased proliferation does not always equate to cystic expansion, let alone initiation of cystogenesis.

Absolutely, we totally agree. We now cite two papers, one from our group, demonstrating that increased proliferation is not sufficient to initiate cyst formation. However, we hope the reviewer agrees that proliferation is considered as necessary for cysts growth and expansion. Thus necessary, but not sufficient would be the definition. We believe this is the general agreement in the field and the reason why targeting proliferation remains the primary therapeutic strategy in this disease. We now try to stress it a bit further.

Therefore, caution should be exercised in making this connection. In addition, it would be informative if they extend the discussion to the role of metabolic reprogramming/metabolic regulation in fibrosis and tubular degeneration/death, in the context of NPH. If metabolic reprogramming has a causal role in NPH, it should affect (partially or entirely) fibrosis and/or cell death, as these are the major drivers of kidney function in NPH. In this regard, and as also mentioned above, the discussion of effects of altered mitochondrial metabolism on proliferation in NPH is somewhat superficial, as the reduced kidney size may be due to increased cell death rather than reduced proliferation.

Thank you. We agree that a deeper discussion on the possible role of metabolic and mitochondrial alterations in renal fibrosis is warranted given the extensive literature on the topic in contexts other than NPH. Thanks for bringing this up. And we agree that apoptosis could play a role as well. Indeed, we were the first to report increased apoptosis in NPH human tissues a while ago (Wodarczyk et al, 2010). We have integrated those points in the discussion. However, we would like to stress here that we are not saying that reduced proliferation could be a driver of NPH (as the increase is not a driver in PKD), as this would imply a developmental defect, the time when proliferation is most active (actually an interesting point to expand in future research and discussions, as this is a recessive disease!). In our view the lack of increased proliferation could justify why kidneys do not become enormous and also why cysts, when present, are rather small. We have tried to specify this better. It is fair to say that, given the limited data available on NPH, most of what one could say is speculative. And this was our intent, even if the few experimental evidences all align with the idea of two opposite roles for cilia. The point we would like to stress here is that the profound differences justify exploring a different interpretation of the common and not common (possibly opposite) paths in the two major ciliopathies known to date, i.e. PKD and NPH. Data from mice, especially the work of Dr. Somlo, now supports a rationale for this interpretation, we believe. Also, we now incorporate the example of rare, but notable, exception to the descriptions above that have been suggested by reviewer 2. Hope the new version reads more balanced and thanks for the critique.

The discussion on primary cilia and metabolic regulation is adequate and informative. Overall, this is a well-done manuscript, which I recommend for publication, after expanding a bit the discussion on fibrosis and tubular atrophy/death.

Thanks, we have done so.

Minor comments:

All minor comments were corrected.

- 1) P.4 line 2, second paragraph: "receptor activation"
- 2) P.6, line 7, second paragraph: "a two-hit disease"
- 3) P.7, line 5, fourth paragraph: "... genes cause misfolding..."
- 4) P.9, line 12, second paragraph: "...axonemal membrane of cilia..." instead of "plasma membrane".
- 5) P.10, lines 17-20, first paragraph: Please add references.
- 6) P.11, line 1, first paragraph. "...plethora of studies have contributed to the delineation of the mechanisms..."
- 7) P.20, line 1, first paragraph, "amino acid..."
- 8) P.20, line 9, first paragraph: "suggest that opposite..."
- 9) P. 18, line 7, first paragraph and P.20, line 1, second paragraph: "...lines of evidence..." instead of "evidences".

Referee #2 (Remarks for Author):

The manuscript by Clerici and Boletta describes recent progress in metabolic programming in polycystic kidney disease and other renal ciliopathies. This is a very well-written manuscript that gives novel insights into a rapidly evolving research field and therefore is of interest for a broader community. It comes timely as recent publications have revealed novel and important aspects. There are a number of aspects remaining that should be addressed prior to consideration for publication.

Thanks, we have addressed all the points.

Major:

- The authors classify ADPKD as a ciliopathy. There is an ongoing debate about this classification. Please consider to rephrase or discuss.

This comment took us a bit off-guard, we must admit. We do not think there is any ongoing debate on PKD not being a ciliopathy. When ADPKD started to be classified as a ciliopathy, in the early 2000s, there was some resistance from the field, indeed. When Greg Pazour showed localization of PC2 in cilia in 2002, technically opening the field of the ciliopathies, the main criticism was directed towards the poor evidence of Polycystin-1 sitting at cilia, due to all major technical issues encountered to immunostain the protein. This is no longer an issue for anyone in the field, as the evidences of cilia being central in ADPKD are overwhelming. It is fair to say that the polycystins are also involved in other subcellular compartments (as most ciliopathies proteins) and this might complicate studies and interpretation. We refer to a recent review written by Dr. Boletta with Dr. Caplan where all this is revised and discussed in detail, with also an historical perspective on the evolution of the evidences. But thanks for your comment, because it brings about the notion of some persistent confusion that we were not aware of, which makes our review even more relevant and timely from this point of view.

- On page 6 the authors speculate about potential opposite ciliary alterations underlying PKD vs. NPH. Please specify why this could not be alterations in other subcellular organelles.

We now introduced a comment.

- The authors state that only one compound was approved for therapy of ADPKD to date. This is not entirely correct. Tolvaptan is the only targeted treatment based on molecular insights but therapy of ADPKD goes far beyond this incl. therapy of arterial hypertension etc. Please rephrase.

Only one compound was approved as specific therapy for ADPKD. Extrarenal manifestations are often treated symptomatically, using anti-hypertensive drugs, painkillers etc. Nothing that impacts on disease progression. Because this is not a clinical review, we now refer to a recent clinical review.

- The authors mention that Glis2 responds to all three major characteristics expected from a CDCA component. They mention that removal of cilia reduces Glis2 activity. Yet, loss of Glis2 results in a NPH-like phenotype and loss of Glis2 in Kif3a-mutant mice results in reduction of the Kif3a-phenotype (Lu et al, Kid Int 2016) - does this fit to the concept of the authors? Can they comment? Thanks for the comment and for bringing up this study that we had entirely overlooked. The study in principle supports what we are summarizing and what Steve Somlo has been working on. However, to know exactly the mechanism and whether these data can be related, one would have to compare the phenotype of *Pkd1* or *Pkd2* mice double mutants for Kif3a and triple mutants for Kif3a and Glis2, to see if there is a gradual improvement (the expected outcome). In all cases, thanks for bringing this up, we now discuss it.

- On page 17 the authors link variants in *Anks6* to cellular metabolism and link this to proliferation. The link remains weak and I understand that causality was not shown. I would support the authors to extend their already cautious wording even further. Can the authors also be more specific here? In the same section it may be important to point out that Glis2 mice show an NPH phenotype. We agree. Please note that the link between *Anks6* and proliferation was done by the authors of the paper, not by our speculation. And thanks for the comment on *glis2*. This is exactly the emerging model and what we would like to bring forward here. *GLIS2* is mutated in NPH. Consistently, *Glis2* mutant mice have an NPH-like phenotype. Removal of *Glis2* improves PKD. This indeed strengthens our proposal that the two diseases might actually be opposite manifestations. To add to the picture, *GLIS2* is an oncogene, whose amplification results in cancer due to massive proliferation. What we are trying to do here is exactly to bring these evidences to light in a single review. We now also cite the *Glis2* mutant mouse that you suggested above, sorry for overlooking. Thanks for your comment.

- The authors speculate that reduced proliferation may be resulting in smaller kidneys. How would the link to fibrosis be in NPH and PKD? Would an additional hit be required? We agree that the fibrosis in NPH is much more prominent, perhaps because the slower growth of the kidneys unveils to a greater extent, while in ADPKD fibrosis comes later. The third hit was hypothesized in PKD where a second genetic hit occurs at the somatic level. NPH being recessive is unlikely to be subject to second or third hits. Nevertheless, ischemic injury could be affecting fibrosis indeed and we now comment.

- Page 18: „BBS is caused by mutations (...) in a complex called the BBSome...". To my knowledge the BBSome includes only some of the proteins that can be affected in BBS. Please double check and comment.
OK, corrected.

- Please use updated nomenclature e.g. „kidney" instead of „renal", „kidney failure" instead of „ESKD" etc. (<https://pubmed.ncbi.nlm.nih.gov/32409237/> or <https://pubmed.ncbi.nlm.nih.gov/32409780/>), „disease-causing variants" instead of „mutations" etc.
We did. Even if this could be debated. As latin-based language speakers, we do not understand why the word “renal”, that came first and that means kidney in latin-based languages should all of a sudden be removed! We have applied in most instances. Thanks for your comment, though.

Minor:

- I would suggest to move the description of the different types of cilia (motile vs non-motile, 9+0 vs

9+2) to an earlier part of the manuscript. On page 4 the authors describe sensing at the node of the developing embryo but have not previously properly introduced motile cilia which could be helpful. Given the limited space we have opted for removing reference to motile cilia and we now only refer to the sensory monocilia.

- On page 7 the authors state that ARPKD „is diagnosed in utero". While this is true in many cases, there are also more and more emerging descriptions of mild courses of ARPKD. Please rephrase.
OK.

- On page 7 the authors cite „Group et al., 2025" - in understand they refer to the KDIGO ADPKD Work group? Please check
Yes, we changed this, thanks

- Kidneys in infantile NPH can be very big during early disease and are a differential diagnosis of ARPKD - please consider your statement on page 8.
Thanks for bringing this up, we now have incorporated the comments and references.

- It remained unclear to me why Tulp was in the figure 2 until reading further down. Please consider to rearrange the text.
TULP3 is actually commented early on in page 9 before figure 2 is cited.

4th Apr 2025

Dear Dr. Boletta,

We are pleased to inform you that your manuscript is accepted for publication and is now being sent to our publisher to be included in the next available issue of EMBO Molecular Medicine.

Your manuscript will be processed for publication by EMBO Press. It will be copy edited and you will receive page proofs prior to publication. You will soon be contacted by Springer Nature to sign your publishing license. When you login to the customer service website, please use the following token to waive the article publication charges. Should you experience any difficulty, please email publishing@embo.org.

Waiver token: XXXXXXXXXXXXXXXXXXXX

Zeljko Durdevic
Senior Editor
EMBO Molecular Medicine